# Integrative functional analysis uncovers metabolic differences between *Candida* species

Neelu Begum[1], Sunjae Lee[1], Theo John Portlock[2], Aize Pellon[1], Shervin Dokht Sadeghi Nasab[1], Jens Nielsen [3,4], Mathias Uhlen [2], David L. Moyes [1✉] & Saeed Shoaie [1,2✉]

*Candida* species are a dominant constituent of the human mycobiome and associated with the development of several diseases. Understanding the *Candida* species metabolism could provide key insights into their ability to cause pathogenesis. Here, we have developed the BioFung database, providing an efficient annotation of protein-encoding genes. Along, with BioFung, using carbohydrate-active enzyme (CAZymes) analysis, we have uncovered core and accessory features across *Candida* species demonstrating plasticity, adaption to the environment and acquired features. We show a greater importance of amino acid metabolism, as functional analysis revealed that all *Candida* species can employ amino acid metabolism. However, metabolomics revealed that only a specific cluster of species (AGAu species—*C. albicans, C. glabrata* and *C. auris*) utilised amino acid metabolism including arginine, cysteine, and methionine metabolism potentially improving their competitive fitness in pathogenesis. We further identified critical metabolic pathways in the AGAu cluster with biomarkers and anti-fungal target potential in the CAZyme profile, polyamine, choline and fatty acid biosynthesis pathways. This study, combining genomic analysis, and validation with gene expression and metabolomics, highlights the metabolic diversity with AGAu species that underlies their remarkable ability to dominate they mycobiome and cause disease.

[1] Centre for Host-Microbiome Interactions, Faculty of Dentistry, Oral & Craniofacial Sciences, King's College London, SE1 9RT London, UK. [2] Science for Life Laboratory, KTH – Royal Institute of Technology, Stockholm SE-171 21, Sweden. [3] Department of Biology and Biological Engineering, Kemivägen 10, Chalmers University of Technology, SE-412 96 Gothenburg, Sweden. [4] BioInnovation Institute, Ole Maaløes Vej 3, DK2200 Copenhagen N, Denmark. ✉email: david.moyes@kcl.ac.uk; saeed.shoaie@kcl.ac.uk

Fungal infections affect around 7.5 million people around the world every year. Within human fungal communities (mycobiome), with the notable exception of the skin, *Candida* species are the most common group[1–3]. These species are generally pathobionts, being the most common human fungal pathogens, despite also being commensal organisms[4]. *Candida* infections are becoming increasingly concerning, and the World Health Organisation (WHO) has recently emphasised international surveillance for diagnosis and management of fungal infection, particularly *Candida albicans* infection[5–7]. Recently, a novel *Candida species*, *Candida auris* has been identified with significant mortality and morbidity, as well as a high degree of anti-fungal resistance[8,9]. There are over 200 *Candida* species currently identified, but only a handful of these are present in the human microbiota with the ability to cause infection and pathology. Most notable examples of these include *C. albicans, C. glabrata, C. dubliniensis, C. tropicalis* and *C. auris*[9–15]. *Candida* species, most notably *C. albicans* and *C. glabrata*, can give rise to a variety of superficial infections, including oral thrush and vulvovaginal candidiasis, but are also capable of causing a systemic infection with significant mortality[16–20]. As well as direct infections, fungi such as *Candida* species have also been associated with oncogenesis through complement activation, demonstrating potential effects of the interaction of fungal species with human host[21].

An essential virulence determinant of fungi is their metabolic plasticity[22]. Fungi are significant in their ability to utilise numerous different anabolic and catabolic sources in their metabolic processes, attributable to switching between carbon and nitrogen sources[23]. Nutritional availability, environmental factors, competition and pathogenic factors all influence this plasticity[24,25]. Investigations of *Candida* species-specific transcriptional regulators of glycolytic genes (e.g. Tye2 and Gal4) and enzymes of the glycolytic pathway (hexose catabolism), indicate these factors play an essential role in central carbon metabolism commonly applied during infection events[22,24,26]. Glycolytic metabolism can activate virulence factors that initiate hypha formation, activate fermentative pathways, repress gluconeogenesis, and the TCA cycle[27–30]. Alternatively, *C. albicans* can switch to gluconeogenesis and the glyoxylate cycle to confer full pathogenesis during systemic candidiasis[31–34]. Carbohydrate metabolism is coupled with changes of cell wall architecture, host immune response modulation, as well as adherence, biofilm formation, stress response and drug resistance[24,35–37]. If carbohydrate sources are limited, *Candida* species can use amino acids and lipids as supplementation for metabolic adaptation[22,38–40]. Amino acids produced by *C. albicans* have been shown to drive tissue damage by initiating stress responses and adjusting the surrounding environmental pH, helping induce of host invasion processes[35,41–47]. Very little is known about the regulation, process and utilisation of amino acid metabolism in *Candida*[39]. However, *C. albicans* is known to use amino acids to replace carbon and other nitrogen sources[48]. *Candida*'s ability to convert arginine to urea allows the neutralisation of an acidic environment triggering the development of hyphae and biofilm formation[32,49,50]. Notably, recent work has shown that *C. albicans* phagocyted by macrophages induces fatty acid β-oxidation and the glyoxylate pathway to induce hypha formation for escape. In a harsh environment that lacks even a nitrogen source, *Candida* can recycle and produce its own proteins and polyamines without host nitrate[51]. Thus, understanding of metabolism and functionality of *Candida* is instrumental in tackling infection and mortality prevalence[52].

Here, we have developed the BioFung tool—a database derived from 128 fungal species using KEGG orthologs (KO) and focused on functional information and interpretation of biological information to address the issue of the lack of resources for functional annotation of fungal genomes. We then go on to apply this database tool to *Candida* species to identify enriched functionality in specific clusters and further show how it can be integrated with other tools such as CAZyme. In doing so, we demonstrate the power of this tool to make functional analyses of fungal species. Distinct clusters of *Candida* species were defined based on literature review of contributions to candidemia and mortality (Fig. 1a), cluster of *C. albicans, C. glabrata* and *C. auris* referred to as AGAu species. This cluster has a high association with infection and mortality, relative to other *Candida* species[18,53–55]. We applied comparative analysis techniques based on gene, protein, and enzyme-substrate levels and identified metabolic pathways in *Candida* species, such as choline and polyamine pathways. Metabolomics validation along with experimental validation from gene expression confirmed important AGAu cluster difference. This study (1) provides a tool for functional annotation of fungal species, (2) highlights amino acid metabolism importance in AGAu species that remarkably dominate the mycobiome, and (3) identifies potential fungal biomarkers and anti-fungal targets in metabolic pathway.

## Results

### Development of BioFung database and functional annotation of *Candida* protein-encoding genes.
There is currently a dearth of tools for accurate and complete annotation of fungal genomes. In order to analyse the global functionality of fungal species, we decided to develop a database tool (BioFung) to help solve this problem. The BioFung database takes a list of protein sequences in fasta format (representing a fungal genome) and outputs KEGG orthologues that are then used to annotate the associated genome, giving an overview of the potential functionality. To perform a global functional analysis of *Candida* species, we collected 49 publicly available genomes of different *Candida* strains covering 13 different species (Supplementary Table 1). We selected species based on their clinical importance and abundance within the human mycobiome[56–60]. All 49 *Candida* strains were isolated from different body sites from people represented in a pie chart demonstrating diverse survival of *Candida* species and in different geographical locations illustrated on a global map (Fig. 1b and Supplementary Fig. 1a). Comparisons of sequencing platform, scaffold assembly and genome were performed to assess how the quality of published genomes impacts on the annotations (Supplementary Figure 1b, Supplementary Table 1). No distinct impacts were seen based on differences in sequencing processes and genome assembly. Although assessment of similarity of sequence across 49 strains is not a direct assessment of integrity of genome annotation, average nucleotide identity (ANI) using nucleotide sequence reveals that the phylogenetic relationship of all these *Candida* species are interlinked (Fig. 1c, Method). We observed strain-specific differences in phylogenetic lineages with 11 distinct branches, including branching of *C. auris, C. glabrata* and *C. albicans*, implying genetic diversity that could implicitly be interpreted into functional variances. To elucidate functional details for these strains, we built fungi-specific Hidden Markov Models (HMM) using fungal gene clusters, named the BioFung database (Fig. 1d, "Method")[61,62]. We analysed 524,288 fungal genes, from 128 fungal species, with a coverage of 4,822 KOs, and 4,430 fungal KO alignments to create BioFung (Supplementary Data 1–3). Comparison of the BioFung database with similar eukaryote-specific HMM sources (Euk90 and Euk100) indicated that BioFung has both higher coverage and specificity of KOs (Supplementary Fig. 2a–c)[61,62]. BioFung was applied to protein sequences of *Aspergillus fumigatus, Aspergillus niger* and *Aspergillus nidulans* for robustness (Supplementary Figure 2d, e).

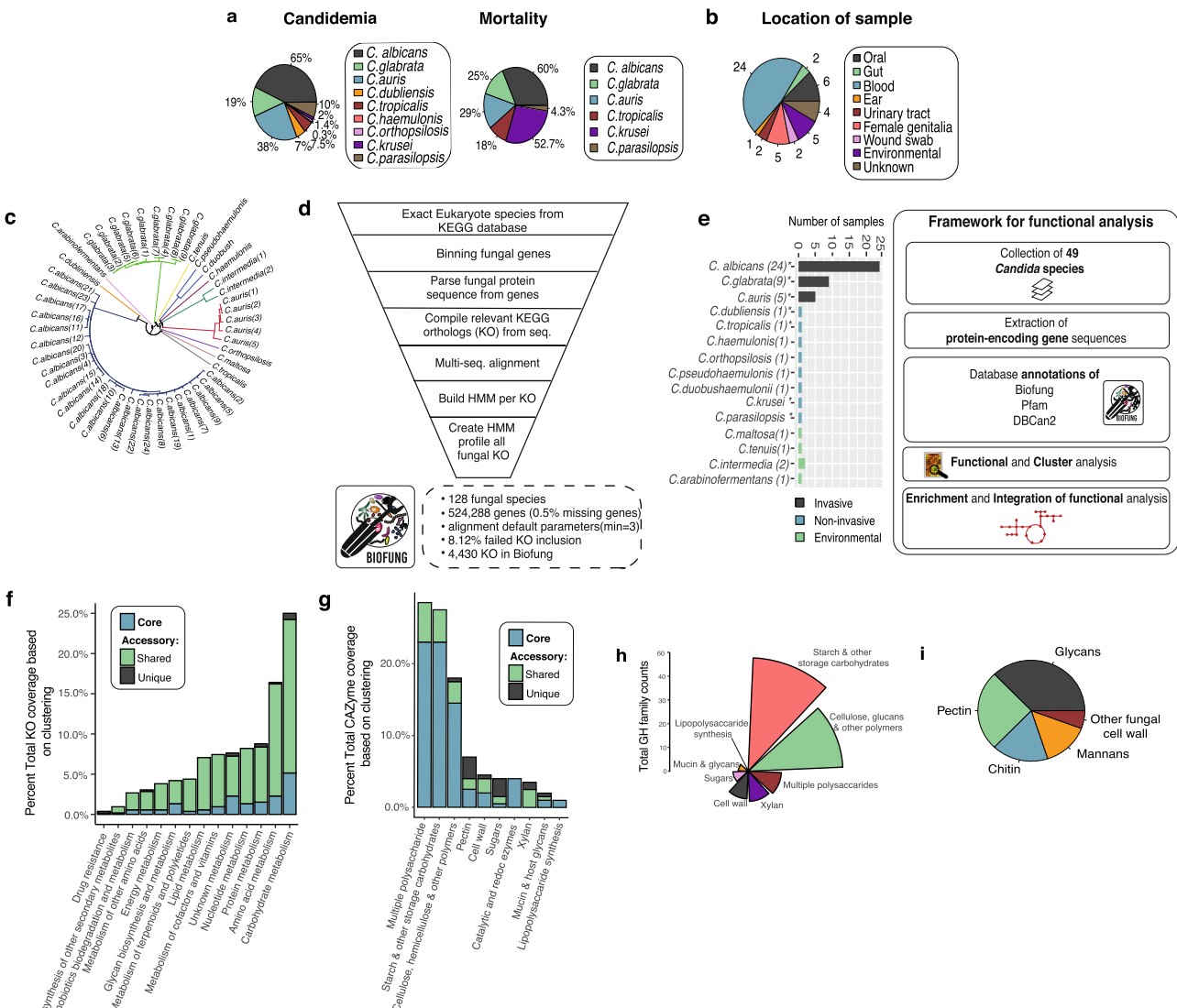

**Fig. 1 Exploration of *Candida* species with BioFung database. a** *Contribution of individual Candida species to candidemia and mortality.* The impact of each species in AGAu species' grouping is attributed in this study (literature-based)[16–18,20,152]. **b** *Candida strain characterisation.* Coverage of *Candida* sample population per species available with the categorisation of species profiled. Numbers around the pie chart signify the number of strain representation in each location. (Supplementary Data 3 for more information about strains and Supplementary Figure 1a for the global representation of *Candida* strains). **c** *Genome-based phylogenetic tree.* The phylogenetic tree was constructed based on average nucleotide identity (ANI) between all strains revealing evolutionary differences across strains (colour coordinated) and indicating distinct metabolic capabilities. See Supplementary Fig. 1b for quality of sequences. **d** *BioFung database creation workflow.* Eukaryote annotation from KEGG database parsed to extract all fungal species. They were genes parsed, sequences extracted and reassembled to KO. The multi-sequence alignment was performed on each KO with all corresponding sequence available. HMM, profile built based on each KO and assembled to provide a more accurate annotation of fungal species for KO. **e** *Distribution of Candida species based on sample collection and the framework of protein-encoded genes analysis of Candida strains.* Strains isolated from the various location providing relevant clinical association to host mycobiome and environmental species. *indicates clinical strains used for metabolomics. Functional analysis performed on 49 *Candida* species collected from public repositories. Protein sequences were annotated with Pfam, dbCAN2 and BioFung database for biological information. **f** *Core and accessory overview of the metabolic pathway across Candida strains.* Shared genome feature refers to 6–48 species sharing the function and unique genome features is exhibited by less than 5 *Candida* species denoting accessory functions. **g** *Clustering of carbohydrate-active enzyme profile (CAZyme).* Core, shared genome (6–48 strains), and unique genome (<5 strains) illustrates distribution analysis of functions across all *Candida* species. **h** *Breakdown of GH family substrate-converter activity.* Analysis of enzyme function of glycoside hydrolase family across all *Candida* strains. **i** *Breakdown of cell wall composition of core Candida strains with identification of 49 CAZymes.*

Output of BioFung is compared to representative organisms in the KEGG database (Supplementary Fig. 2f).

The collection of *Candida* strains used to integrate functional annotations can be categorised into commonly invasive and non-invasive (requiring a secondary factor to cause infection, such as co-morbidity, immunodeficiency) based on the literature (Fig. 1e, Supplementary Table 2, "Method"). These representative samples

of *Candida* were integrated into the functional analysis framework, with a total of 49 *Candida* species annotated with BioFung, Protein families (Pfam)[63] and Carbohydrate-Active enZyme (CAZyme)[64] databases. We applied BioFung using the UCLUST algorithm, to establish core genome features (found in all *Candida* species) and accessory genome features (shared or unique functions)[65]. In covering KEGG metabolic orthologs,

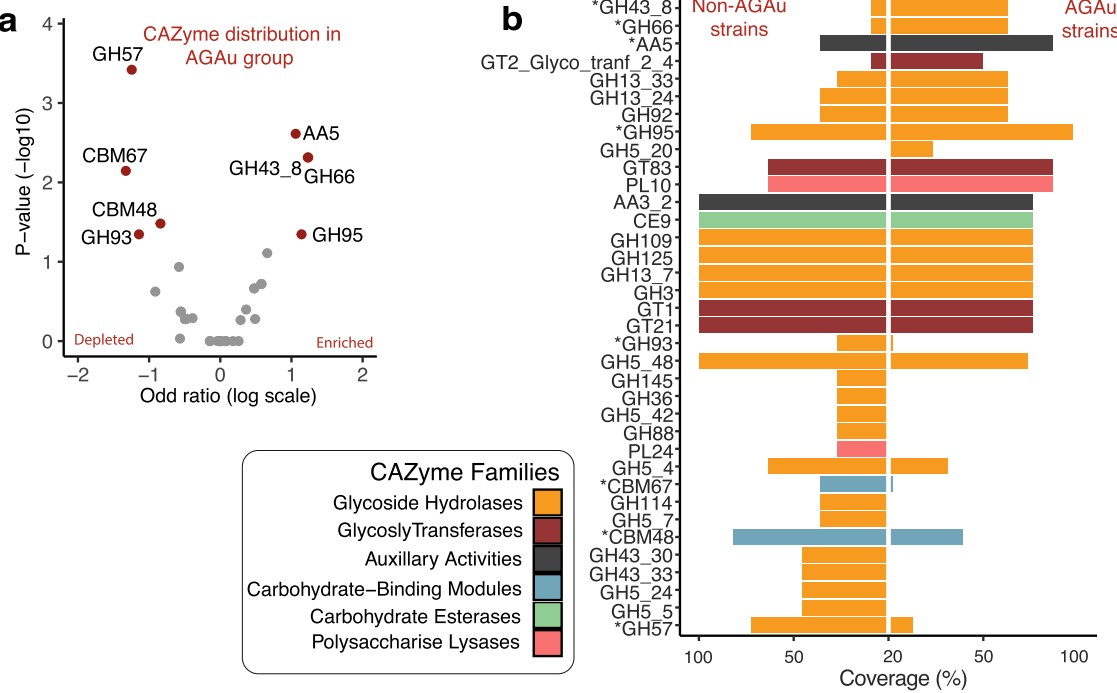

**Fig. 2 CAZyme profile in AGAu *Candida* strains. a** *CAZyme analysis across AGAu and non- AGAu groups.* (*Asterisk connotes statistically significant CAZyme, Chi-square tests, p values < 0.05).* **b** *CAZyme enrichment and depletion in AGAu strains.* Volcano plot showing statistically significant evidence of a relationship between AGAu strains (Chi-square tests, p values < 0.05 and odds ratio on enriched and depleted in AGAu strains, in red).

clustering analysis determined a larger number of accessory features of metabolism compared to core characteristics seen in all *Candida* strains (Fig. 1f, Supplementary Fig. 2g–i, "Method"). Intra-strain analysis of *C. albicans* across 24 strains sequenced showed largely conserved metabolic pathways and CAZyme profiles (Supplementary Fig. 2j, k).

**Identification of global functional annotation profiles in *Candida*.** We next determined the CAZyme profile by mapping the 49 *Candida* protein sequences to the dbcan2 database[64]. Doing this allowed us to infer molecular enzyme function[64]. CAZymes are vital enzymes involved in the metabolism of complex carbohydrates. Approximately 205 unique CAZymes were identified in all *Candida* strains, with various functions (Fig. 1g, Supplementary Data 4, "Method"). From core *Candida* genome analysis, annotated enzymes were distributed across 6 active CAZyme families, with an assortment of enzymatic functions. The glycoside hydrolase (GH) family showed the highest degree of core coverage (Supplementary Fig. 3a), with much of the GH family activity in starch and other storage carbohydrates substrate converters (Fig. 1h).

Previous reports have determined the importance of fungal cell wall composition a crucial virulence factor during infection, and assessing CAZyme components of fungal cell wall substrate converters has been extensively researched[66]. Here, we reveal the presence of pectin lyases, glycan lyases, chitin lyases and mannan lyases (Fig. 1i). Xylan and sugar utilisation appears to be surprisingly present in the accessory genome (Supplementary Data 4). Pectin substrate-conversion enzyme has been identified as the core feature of *Candida* species' functional cell wall enzyme, though previously only reported in *Candida bodinii*[67] and frequently seen in the fungal plant pathogen, including *Aspergillus Pencillium*[68]. Alongside ß-glucan, mannan and chitin carbohydrate enzyme profiles, *Candida* cell wall activity includes pectin enzyme activity (Supplementary Table 3).

In addition, we identified 1182 Pfam clans from all *Candida* strains and re-categorised them into 14 functional clans (Supplementary Data 5). Pfam domain annotation indicating genetic information processing, cell machinery, and metabolism was among the most extensive Pfam domains exhibited (Supplementary Fig. 3b). We assessed the diverse functional association of protein domains by analysing core functional clans, and determined similar patterns of dominance for carbohydrate, amino acid and lipid processing-associated domains (Supplementary Fig. 3c).

**The functional and metabolomic activity of clinical AGAu *Candida* strains.** Next, to better explore and understand the link to metabolism and pathogenesis, we clustered *Candida* species into groups based on the invasive nature of particular species, from literature search of species contribution to candidemia and mortality (Fig. 1a). *C. albicans, C. glabrata* and the emerging invasive species *C. auris* were grouped together (AGAu cluster). Alternative *Candida* species termed non-AGAu group include opportunistic species that require virulence factors or a defective immune system to cause disease pathology as well as environmental *Candida* species. The AGAu cluster contains those *Candida* species most commonly associated with clinical pathology, contributing to a higher percentage of mortality and candidemia[16–18,53–55,69]. This classification of AGAu is analysed and discussed throughout the rest of this paper.

We compared the CAZyme profile coverage of AGAu and non-AGAu groups, with classes highlighted based on colour (Fig. 2a, b, Supplementary Table 4). The CAZyme GH43_8 (substrate conversion of α-L-arabinofuranosidase/β-xylosidase[70]) was significantly enriched in AGAu possibly involved in the breakdown of complex glucans (Wilcoxon rank-sum test, $P < 0.05$)[71]. The identification of significant CAZymes in the AGAu cluster showed carbohydrate conversion of xylan (GH43_8), mucin (GH95), cellulose (GH66) and copper oxidase family (AA5)[72–74]. The presence of xylan substrate converter is unanticipated as xylan is only commonly found in plant cell walls

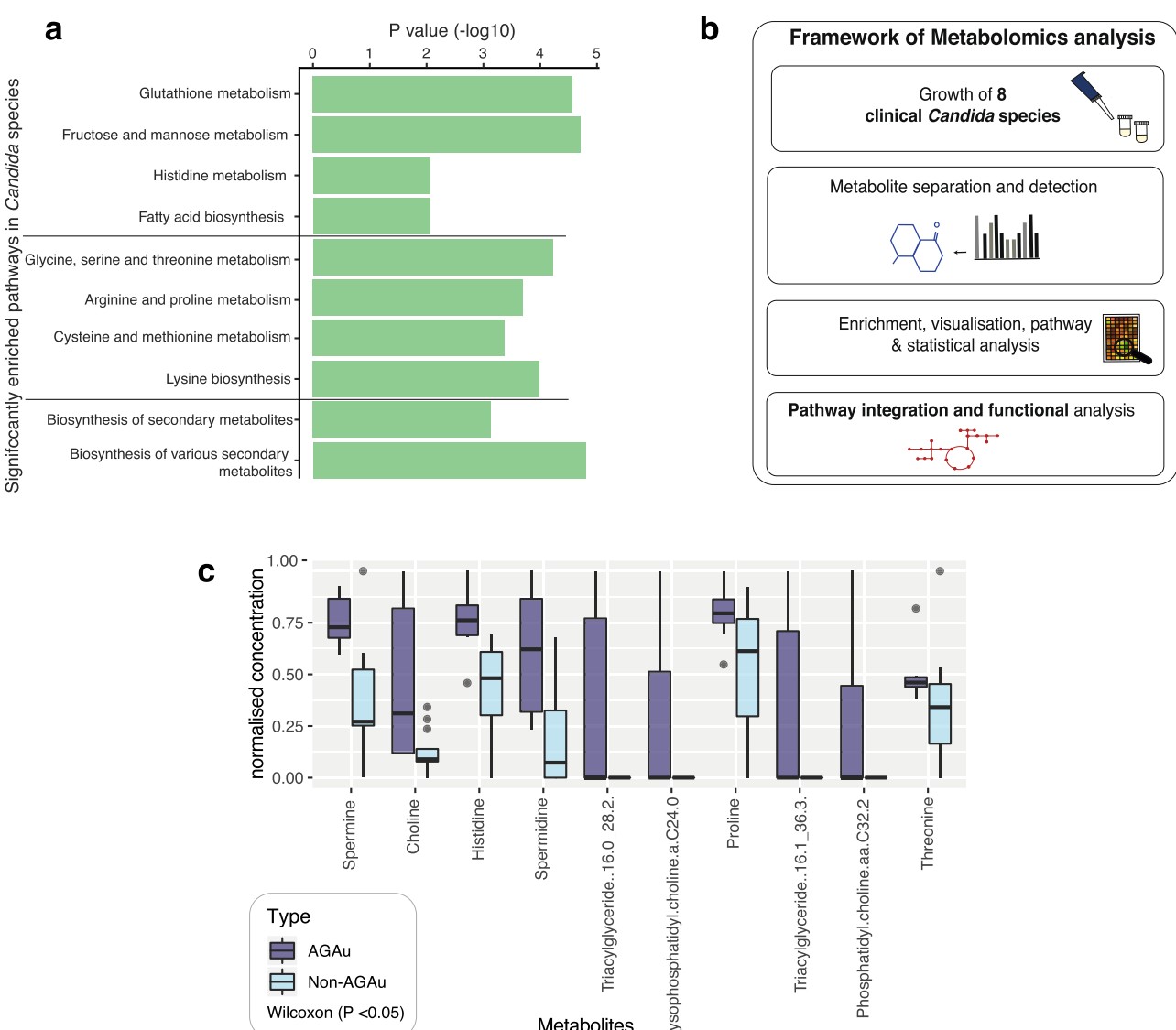

**Fig. 3 Comparison of functional and metabolomics in AGAu and non-AGAu strains. a** *Pathway enrichment in 49 Candida strains.* Significant metabolic pathways highlighted in *Candida* species indicating genetic pathway potential (Hypergeometric tests on contrasted annotation between AGAu and non-AGAu species, *p* values < 0.05). **b** *Framework outline of metabolomic analysis of Candida species' metabolism levels of bioactive metabolites in Candida strains exhausted media.* Computer simulations were performed for pathway analysis, and statistical approach was applied for candidate metabolites that have a potential influence on the host. **c** Enriched metabolites detected in targeted metabolomics between AGAu and non- AGAu groups (Wilcoxon rank-sum tests, *P* values < 0.05, middle line for median with interquartile range (IQR) and whiskers 1.5× IQR, "Method").

and more likely present in non-AGAu species. The presence of GH43_8 has not been associated with any other host, bacterial or fungal species. Interestingly, GH66 associated with human oral plaque formation[72] and AA5 enzyme has been reportedly linked to fungal defence[75]. CAZymes seen in non-AGAu species showed carbohydrate-binding module families involved in sugar, poly-saccharide and cell wall breakdown, including CBM48.

The AGAu species are morphologically diverse, (*C. albicans* is dimorphic—capable of growing a both filamentous hyphae and single-celled yeast, whilst *C. glabrata* and *C. auris* are yeasts), and they are all potential pathogens. Based on contrasting KO annotations, the genetic potential of pathways between AGAu and non-AGAu *Candida* strains showed little difference with hypergeometric testing. Nevertheless, there is functional evidence of significance in these pathways present in both clusters, indicating a genetic potential for all *Candida* strains to undertake these metabolic trajectories (Wilcoxon rank-sum test, *P* < 0.05; Fig. 3a). All *Candida* strains notably revealed encoded pathways facilitating

carbohydrate catabolism within the system; thus, demonstrating the potential to drive increasing metabolic activity, for example through fructose and mannose metabolism. We did, however, identify significant enrichment of amino acid metabolism with metabolomics highlighting trajectories for arginine, proline, cysteine and methionine metabolism. We also observed significant levels of fatty acid biosynthesis and glutathione metabolism, which have previously been associated with virulence mechanisms[76–79].

**Metabolomics revealed key metabolic pathways assimilated by AGAu group.** To elucidate the metabolic trajectory taking place by each cluster group, we performed metabolomics on a collection of 7 clinical *Candida* isolates, representing the diverse pathogenic species (Fig. 3b, "Method"). These clinical samples were previously isolated from patients infections[53,80–83] and were used to evaluate in vitro the critical metabolic activity predicted by our functional analyses (Supplementary Fig. 4a). These

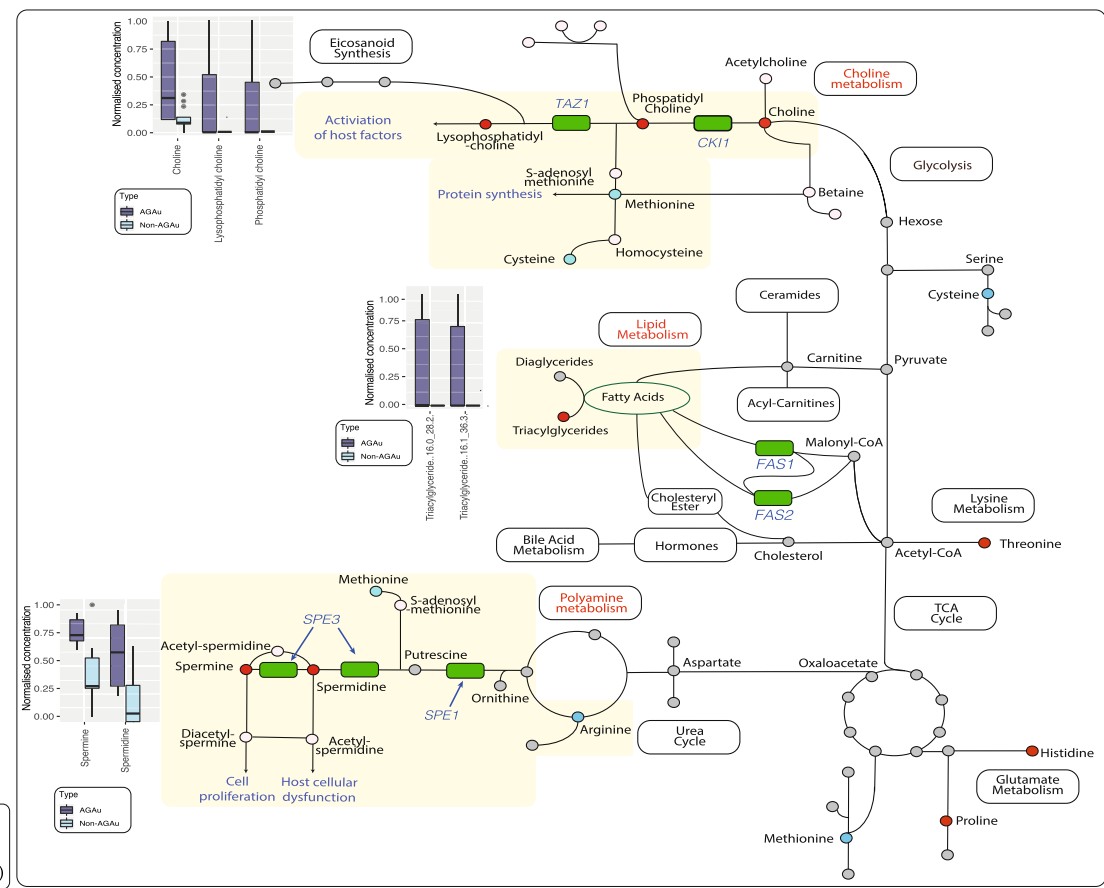

**Fig. 4 Global map metabolism in AGAu *Candida* strains.** Enriched pathways observed in functional annotation and further validated by metabolomics with key metabolites markedly significant. The boxplots indicate metabolomics of individual metabolites in corresponding pathway (Wilcoxon rank-sum test, $P$ value < 0.05, median middle line IQR and whiskers 1.5× IQR). *Colour in pathway red identify significant metabolites, blue indicate significant pathway enrichment and green denotes validation with gene expression.

*Candida* isolates were representative of both AGAu and non-AGAu grouped species. The metabolomics data were used for partial least squares discriminant analysis (PLS-DA), which resulted in distinct cluster separation of significant analyte classes (Supplementary Fig. 4b–d, "Method"). The PLS-DA models identified distinct metabolites feature in AGAu *Candida* species (Supplementary Fig. 4e).

For example, histidine metabolite production in AGAu species supports this metabolite role in systemic infection and is potential an anti-fungal target[84,85]. We also identified choline-derived metabolites (choline, phosphatidylcholine and lysophosphatidylcholine) as increased in certain AGAu species (Fig. 3c; Wilcoxon $P < 0.05$). We identified phosphatidylcholines analyte class as contributing the highest number of features across selected clinical *Candida* species (Supplementary Fig. 4d), which has been observed previously in the hypervirulent *C. albicans* (SC5314) strain[38–41]. Spermine and spermidine were found to be significantly associated with AGAu strains, indicating polyamine metabolism could play a functional role in the increased association with disease pathology of these strains.

**Integrative global metabolic map of AGAu *Candida* species.** Having identified these pathways in silico, we next determined gene expression levels of essential polyamine (SPE11, SPE3), choline (CKI1, TAZ1) and fatty acid (FAS1, FAS2) pathways in *C. albicans* (representative of the AGAu cluster) to validate our findings (Supplementary Fig. 5a, Supplementary Table 5 and "Methods"). All 5 genes showed expression, demonstrating the

activity of these pathways. Our conclusions of key pathway associations draws importance of nitrogen sources, specifically the metabolism of amino acids, in this process (Fig. 4), although to date, *Candida* species pathogenesis is better known to be driven by carbon sources[22,37,86,87]. Increased levels of metabolites were exhibited in AGAu species in the choline pathway, polyamine and fatty acid biosynthesis pathways that are primarily propagated through arginine, cysteine, and methionine pathways (Supplementary Fig. 5b–d). Based on integration of computational and experimental data revealed fundamental metabolic pathways applied AGAu species providing a link to the major advantage shown by AGAu species across the human body with considerable contributions to pathogenesis. These important pathways include polyamine, choline and fatty acid biosynthesis. For instance, the polyamine pathway is thought to be involved in *Candida* cell proliferation, and in turn, causes host cellular dysfunction by modulating acetylation levels of aminopropyl groups and inducing autophagy, thus increasing cell life span[86,88,89]. We observed fatty acid biosynthesis production with large numbers of metabolites of triglycerides featuring in PLS-DA and family have previously been reported to promote germination and virulence of AGAu *Candida* strains (Supplementary Fig. 4d)[90,91]. Further, fatty acid biosynthesis is vital in fungal cell membrane viability, energy storage, signalling, and cell proliferation—all functions critical in pathogenesis[92–95].

## Discussion

In this study, we develop a tool for analysing fungal metabolism—the BioFung database. Here, BioFung was used for building

metabolic maps of key *Candida* strains and the database provides the mycology community with a resource allowing them to dive more deeply into all fungal species' metabolic capability based on protein-encoding genes. Development of data generation technologies development, tools and database for fungal species is currently in its infancy, despite significant advances in these areas for bacteria and archaea. This database enables detailed mechanistic annotations to optimise our understanding of fungal species. to do this, it uses HMM to provide high specificity for fungal annotation. As such, it is currently the best database available for KEGG-based functional annotation of fungi (Supplementary Fig. 1a–c and Supplementary Data 1–3). Currently, alternative KEGG annotation tools are available including Figfam[96], Pathogen-Host Interaction[97], GO terms[98], Blast2Go[99] and InterPro[100]. However, none of these provides fungal-specific pathway annotation. BioFung is reliant on the KEGG database and current fungal genome annotations, but in doing so is specifically oriented to fungi. Moreover, as updates to KEGG and advances in the functional annotation of fungal genomes become available, they will increase the power of this tool. The robustness and effectiveness of functional annotation of BioFung compared to similar tools have been explored in Supplementary Figure 1d–f, demonstrating the utility of this tool.

Analysis of annotations allowed us to identify the influential AGAu group of *Candida* strains, highlighting critical metabolic pathways in these strains. In doing so, we developed increased understanding of the metabolism of these strains through integrating multi-omics and experimental data. BioFung can be used extensively to better understand individual fungal species' metabolic pathways but can be extended to explore metabolic interactions between fungi, other organisms, and within the host mycobiome.

Using BioFung in combination with metabolomics validation indicates that the AGAu species appear to be employing specific pathways in amino acid metabolism to potentially improve their competitive fitness during host pathogenesis. This shows a degree of metabolic plasticity indicative of fungi, where secretion of these metabolites associated with these pathways aiding in better adaptability to growth, virulence factor production, hyphae and biofilm formation[39,101], enabling more effective adaptation to a wide variety of environments and habitats. Amino acid metabolism has been proposed as an alternative energy source in stress responses and an alternative to carbon sources for growth. We demonstrate here that all *Candida* species have an amino acid pathways to employ metabolic remodelling (Fig. 3a)[39,42,86]. However, integrating metabolomics from strains grown with abundant nutrient source, shows that the AGAu group have significantly more active production of polyamine and choline metabolites, which requires the use of amino acid metabolism for production (Fig. 4). The AGAu group employs arginine, methionine and cysteine metabolism and more extensive exploration and experimental data needed to understand the causal effect of amino acid metabolism. For instance, we have identified a confirmed target pathway for anti-fungal drugs with glutathione metabolism (GSH), attributed to fungal mitochondrial maintenance, preservation of membrane integrity, regulation of transcription factors in stress response and protection against reactive oxygen species. Reducing activity of GSH is under investigation as supplementary aid for anti-fungal drugs (azoles and echinocandin) against resistant strains[102–105]. This finding verifies that pathway enrichment analysis echoes feasibility in clinical relevance within the host.

Among these metabolites identified are the polyamines, including spermine and spermidine. Polyamines play critical roles in normal cell physiology. Spermine is essential for *Candida* hyphal formation, playing a pivotal role in *Candida* invasion[106].

Spermidine, meanwhile, drives genetic modification in fungi by regulating cell cycle and translating the modification of eukaryotic initiation factor (eIF)[107–109]. Excessive polyamines prolong yeast survival via delayed DNA degradation, increasing the likelihood of mutations that could contribute to the development of anti-fungal resistance[110,111]. These mutations are an important consideration, given that *C. glabrata* and *C. auris* are heavily associated with rising anti-fungal resistance[112–114]. Polyamines have also been shown to be anti-inflammatory, depending on the microenvironment, potentially explaining the additional benefits of secondary metabolites to *Candida* species by modulating host immune responses[115]. The use of polyamines is not limited to fungi. Bacteria use polyamines to create and maintain biofilms in order to withstand host defences as well as promoting cancers[116–118]. Viruses use polyamines to promote cell proliferation, thereby promoting their propagation and spread. Intervention in polyamine synthesis has a high degree of potential as a target for antimicrobials. DNA viruses upregulate polyamine synthesis in host during infection and blocking polyamine synthesis is a strategy used in broad-spectrum anti-viral[119–121]. These examples along with our findings here indicate that manipulating polyamine secretion from *Candida* species is a realistic target for therapeutic intervention of associated diseases.

Choline metabolism is a critical function for both microbial and host physiology, as demonstrated by the increase seen in AGAu *Candida* species' related metabolites. Disruption of phospholipid biosynthesis in fungi can occur through inhibition of phosphatidylcholine synthesis, showing preventing virulence within the systemic mice model[122–125]. Further, acetylcholine is essential in the formation of the chitin wall characteristic of fungi[126,127]. Along with the bacteriome, *Candida* species contribute towards host acquisition of choline. As understanding of choline metabolism is in its infancy, further investigation of host–mycobiome interactions is needed, potentially providing insights for repurposing potential therapeutic intervention. For instance, lack of choline in humans drives liver dysfunction due to the accumulation of lipids within hepatocytes, which can lead to fatty liver diseases and even hepatic liver cancer[128–132]. We acknowledge the limitation of using in vitro metabolomics experiments as the carbon source of the microbial media used (SAB) is not representative of the profile of sources available during human host infection. However, use of these metabolomics datasets does give an indication of potential differences in the metabolic potential of different *Candida* species, indicating that this approach would have real value in exploring clinical metabolomics datasets from different fungal infections.

Functional analysis indicates that both the AGAu and non-AGAu groups show a high degree of metabolic plasticity (Fig. 3a)[56,59,60,133]. CAZymes also demonstrated functional differences seen between AGAu and non-AGAu groups. The unexpected findings of xylan in AGAu cluster, that is normally present in plant cell walls. We contemplate that AGAu clusters are pathobionts in humans, they are also common environmental fungi, and thus the enrichment of functions related to breakdown of wood biomass potentially reflect the range of environments and nutrient sources that can be utilised by the AGAu cluster. It is possible that this is a further reflection of their virulence, demonstrating that one of their main virulence attributes is an ability to thrive in a variety of environments.

The AGAu group show GH66, which has previously only been associated with the human oral microbiome and as a potential marker for plaque formation[72]. Given that *C. albicans* is a constituent of oral plaque, this is consistent with clinical data (Fig. 2a, b)[134]. The function of GH43_8 found in the AGAu group is inconclusive but was recently detected in bacteria as β-galactofuranosidase[70]. Although the modes of action for both

GH43_8 and AA5 are currently unknown their enrichment in the AGAu group may provide a function-targeted biomarker for *Candida* infections[135,136]. We also highlight fatty acid biosynthesis pathway in AGAu species with significant levels of triglycerides production detected (Fig. 3c). Fatty acid synthesis has been identified in *Candida* species previously, with focus on *OLE1*, *FAS1* & *FAS2* genes as key indicators to pathogenesis and virulence[76–79]. This validates the notion of targeting fatty acid biosynthesis pathway within *Candida* species to disrupt *Candida* overgrowth in the host.

Our study has addressed the need for functional data and tools for fungal species by developing the BioFung resource using the KEGG database. This enables detailed mechanistic pathway analysis of fungi. Our integrative analysis of the AGAu group (associated with the disease pathogenesis) highlighted key pathways that potentially increase virulence and have associated effects in the host. We hypothesise that these markers can aid in identifying routes for intervention in invasive infection and suggest polyamine, choline and fatty acid biosynthesis metabolism as potential targets for further investigation. The presence of these metabolites from AGAu *Candida* species potentially directly affects host homoeostasis with the mycobiome and adversely affects the host during infection. As such, the AGAu *Candida* species' metabolic reprogramming may present a method of controlling interaction and infection with these fungi. Finally, we focus on fungal metabolism exploration and distinctively towards amino acid metabolism, playing a more significant role in virulence and pathogenicity.

## Methods

**BioFung database construction**. Kyoto Encyclopedia of Genes and Genomes (KEGG) database were downloaded for the investigation of all 128 fungal species (3GB file size) from eukaryote database (5GB file size) (downloaded on August 2019)[137]. Around 1,210,746 genes, which are annotated with 4717 KEGG orthology (KO), were selected among 128 fungal species genes. There were 6071 fungal genes missing sequence to place into KO, and 105 KO failed in multi-sequence alignment due to default settings (minimum of 3 genes sequence required). Those genes per each KO were performed multiple-sequence alignment by ClustalW and generated Hidden Markov Model (HMM) profiles using the hmm-build function of HMMER software (Fig. 1A for workflow and supplementary 2a for coverage)[138,139]. BioFung database is a fungal-specific HMM model made up of 4,722 KOs was freely shared via Github repository (https://github.com/sysbiomelab/BioFung). Missing KO from fungal species was not added due to missing gene sequences from KEGG, or the low number of sequences per KO (<3), thereby failed to perform ClustalW alignment (See details in Supplementary Data 3).

**BioFung quality assessment**. Quality check was performed by comparing BioFung HMMs to pre-trained HMMS for eukaryotes (euk90 and euk100 version 91.0) from Raven Toolbox[57,58], and we observed that BioFung coverage was much higher than other eukaryote profiles (Supplementary Fig. 1c). Application of BioFung to *Aspergillus species* present in KEGG database were compared to assess robustness (Supplementary Fig. 2f).

**Application of BioFung and other functional annotations**. Fungal KO annotation of each species was performed by HMM scanning of BioFung HMM models by HMMER software. An in-depth exploratory analysis was performed by manually checking KO annotations of individual species. Pathway abundance for AGAu and non-AGAu species was performed using KEGG pathway annotations. Hypergeometric testing uses hypergeometric distribution for pathways and computing *p* value with Wilcoxon rank-sum test (<0.05). CAZyme annotations were performed by mapping *Candida* protein sequences using HMMs of dbCAN2 database[64]. Substrate conversion of CAZyme families was checked based on literature review[68,140–146]. *Candida* protein sequences to map against Pfam-A families using HMMs, that are fully annotated and curated above a threshold[63]. Pfam clans' annotations were sub-set into a broader annotation based on a reported standard function of protein domains (please see Supplementary Data 5).

**Genome sequence collection**. Genome sequences of 49 *Candida* species were collected from NCBI database with version release 45 of Ensemble Fungi (date accessed: April 2019)[147,148]. Applied assembly strategy and sequencing platform is associated to predicting gene function, thus supplementary information of sample strain, genome ID, ENA ID, Biosample ID, sequence platform, year of collection,

sample location of collection, sample tuple and available biological annotation has been provided in Supplementary Table 1. The quality of the sequences was checked to look at GC content, scaffold and genome size (see Supplementary Fig. 1b). Nucleotide sequences were used with Average Nucleotide Identity (ANI) to determine the phylogenetic relationship and determine differences between strains using Pyani package[149].

**Contrasted functional annotation of** *Candida* **species grouping**. Presence and absence of microbial annotations, i.e., prevalence, was tested for significance based on condition using Chi-squared tests and odd ratio. Percentage coverage of each was also tested between AGAu and non-AGAu *Candida* species (Supplementary Data 6). Contrasted functional annotations were checked on individual strains and placed into presence/absence to perform chi-squared for significance (<0.05), and the odds ratio was performed to identify enriched and depleted in AGAu samples. Additional significant functional annotations are seen in AGAu cluster (Supplementary Table 4).

**Clustering of protein sequences**. Core, shared, and unique proteins were identified based on sequence similarity by a clustering approach called UCLUST algorithm[65]. In short, UCLUST algorithm was applied to identify similarity in protein-encoding gene sequences by clustering and unique protein sequences were identified if included in singleton protein clusters. Core proteins were identified if corresponding proteins from all 49 species were included in the same cluster. Shared proteins were selected if they did not belong to unique and core proteins. Protein sequence clusters were selected based on a threshold 0.5 for representative seed sequence, a default threshold in UCLUST software[65] (Supplementary Fig. 2g). Based on definitions of the core, shared, and unique proteins, we were able to determine the core, shared and unique annotations for KO and CAZymes, accordingly.

**Strain growth**. 8 strains of *Candida* species (*C. albicans* (SC5314), *C. dubliniensis* (CD36), *C. tropicalis* (CBS94), *C. glabrata* (CBS 138), *C. auris* (47477), *C. parapsilosis* (73/037), and *C. krusei* (CBS573)). Strains were grown in 5ml liquid sabouraud dextrose broth (Thermo Scientific-Oxoid microbiology, UK)[150]. All strains were grown in a shaking incubator 95rpm at the temperature of 25 °C. Timepoint measurement of growth was taken to measure the exponential and stationary phase of the optical density of 1 at 600 nm absorbance (iEMS Ascent absorbance 96-well plate reader).

**Collection and targeted metabolomics on fungal extracellular matrix**. Mid-exponential phase indicates bioactive metabolites and time points for the extraction of extracellular metabolites (see Supplementary Figure 4a). Five hundred microlitres of extracellular medium, proximity to the pellet was removed from growing fungal cells. Samples were placed through a 20 μm Whatman filter and snap-frozen in liquid nitrogen. Targeted metabolomics performed using the MxP Quant500 kit (Biocrates, Austria). Partial Least Square—Discriminant Analysis (PLS-DA) was performed on targeted metabolomics of fungal extracellular matrices and media as control, using *ropls* package[151]. First, PLS-DA was performed to distinguish between *Candida* samples and control (media). Further, PLS-DA was performed to distinguish between AGAu species and non-AGAu *Candida* samples. PLS-DA indicated a significant difference between AGAu clusters (Supplementary Data 7). Further analysis of metabolite concentrations of targeted metabolomics was normalised, and the Wilcoxon rank-sum test was performed to identify critical metabolites (<0.05). Boxplot mid-line notes median with inter-quartile range (IQR) with whiskers 1.5x IQR.

**Validation experiment**. RNA was extracted from 3 biological repeats *C. albicans* (SC5139) using RNA Qiagen Powersoil kit adapted with bead beating with interval placement on dry ice and additional 100 μl of isopropanol. DNAse clean-up performed using RNA clean-up and concentration kit (NORGEN, Biotek corporation). Primers designed for specific amplification of genes SPE1 targeting Ornithine Carboxylase, SPE3 gene for spermidine synthase, CKI1 specific for bifunctional choline kinase/ethanolamine kinase, TAZ1 gene focused on lysophosphatidylcholine acyltransferase and FAS1/FAS2 gene target for fatty acid synthase (Supplementary Table 5 for primer information). These primers are specific for *C. albicans*. Other *Candida* species only predicted gene ontology-based on *C. albicans* and *Saccharomyces* annotation (http://www.candidagenome.org/cgi-bin/GO/goAnnotation.pl?dbid=CAL0000224407&seq_source=C.%20auris%20B8441). Conventional RT-qPCR was performed to identify the expression of these critical pathways for samples, two standard curve analysis with RDN25 which encodes the 25s rRNA subunit and error bars are representative of mean ± SD.

**Statistics and reproducibility**. All statistical analyses were performed using R software v3.6.3. In the analysis for functional pathway annotation with BioFung hypergeometric testing uses hypergeometric distribution and *p* value computed with Wilcoxon rank-sum test (<0.05) using built into R version 3.6.2 package. We further conducted comparison of species with CAZyme annotation by applying chi-squared (<0.05) to determine significant CAZyme and odd ratio analysis indicated presence and absence of these CAZyme in AGAu species. To test

significant metabolites between AGAu and non-AGAu species, Wilcoxon rank-sum test (≤0.05) was performed on normalised concentration of significant critical metabolites.

Strains of *Candida* species analysis was based on availability of protein sequence at the time of data collection at NCBI database. Metabolomics was performed for triplicate biological experiments and placed through MaxQuant500 kit that was analysed with PLS-DA. qPCR validation with *C. albicans* was performed for each primer in triplicates

**Reporting summary**. Further information on research design is available in the Nature Research Reporting Summary linked to this article.

## Data availability

BioFung is public open access database that can be downloaded from GitHub repository: https://github.com/sysbiomelab/BioFung.

## Code availability

The instruction and the pipeline scripts for BioFung can be found at our GitHub repository https://github.com/sysbiomelab/BioFung. BioFung.hmm file was uploaded via GitHub large file storage (lfs) on the GitHub page. Additionally, BioFung is on an automated pipeline with Nextflow v21.04.1 and either singularity v3.8.3 or docker 20.10.7. All software executed by the pipeline is containerised meaning that no additional installation is required for both local or high-performance computing. ReadME file contains code for usage and example fasta.

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

## Acknowledgements

This study was supported by Engineering and Physical Sciences Research Council (EPSRC), EP/S001301/1, Biotechnology Biological Sciences Research Council (BBSRC) BB/S016899/1, Science for Life Laboratory, Swedish National Infrastructure for Computing at SNIC through Uppsala Multidisciplinary Centre for Advanced Computational Science (UPPMAX) under projects SNIC 2020-5-222, SNIC 2019/3-226, SNIC 2020/6-153, SNIC 2021/5-248, SNIC 2022/5-334 and King's College London computational infrastructure facility, Rosalind (https://rosalind.kcl.ac.uk) for high-performance computing. We thank Professor Bernhard Hube for kindly sending *C. parapsilosis* strain.

## Author contributions

N.B, D.M. and S.S. conceived the project. N.B. performed all sample preparation, metagenomics, metabolomics, gene expression data preparation and extraction protocols for the paper. N.B. developed the pipeline, analysis, and made all the figures. N.B. wrote and drafted the figures and manuscript. S.L. advised on design, statistical and functional analysis of the data. T.J.P advised on streamlining BioFung pipeline. A.P. and S.S.N. processed prepared samples on qPCR platform. J.N. and M.U. provided critical feedback on the data and manuscript. All authors read, edited, and reviewed the manuscript.

## Competing interests

The authors declare no competing interests.
