## [Peer Review File · Communications Biology]

Reviewers' comments:

Reviewer #1 (Remarks to the Author):

See attachment

Reviewer's comments:

Manuscript titled "Integrative functional analysis uncovers metabolic differences between Candida species."

The study is interesting as the authors, Begum N et al. have developed the BioFung database – a fungal specific tool for functional annotation using the KEGG database that provides an efficient method for annotation of protein-encoding gene which will be helpful for understanding the functional pathways of the mycobiome. The study focuses on the metabolic diversity within AGAu species of Candida that underlies their remarkable ability to dominate the mycobiome and cause disease.

However, there are certain restraints in this study.

1. It is a bit unclear to understand whether the authors are trying to emphasize on the new BioFung tool or the functional role of Candida species in humans.
2. The authors claim to have 128 fungal species in their database and developing the tool to elucidate the functional role of fungi; therefore, it would have been ideal to have shown the robustness of their tool using different fungal taxa, e.g., *Aspergillus*, *Malassezia* etc. and not just focus on one taxon Candida using 8 different species.
3. The authors should mention the limitations of their study and the benefits of their tool over other similar bioinformatic tools available as an open source.
4. The abstract should describe full forms of the terminologies before using abbreviations. e.g., "KEGG", AGAu species.
5. The methods section needs to be streamlined and overall, the manuscript requires major revision in presenting the findings with emphasis on the BioFung tool.
6. The references section needs to conform with the journal guidelines – the organisms names should be italicized, some text is in uppercase where not required, few references missing the information on Journal volume and page numbers.
7. Overall, the manuscript needs and the results require detailed statistical analysis.

Reviewer #2 (Remarks to the Author):

This manuscript describes the assembly of a Candida yeast specific database and the use of this

database to support

an attempt to separate specific pathogenic strains. The main novelty comes from an analysis of extracellular metabolites where spermine and spermidine metabolism appears to be associated with invasiveness.

Major points

This manuscript is unfortunately laden with many small errors. This manuscript needs proofreading and the attention of someone proficient in English. I do recommend spelling the authors' names correctly.

The Spelling errors that I found are marked in red in the pdf file.

It is not clear to me how the BioFung database is being made available to the readers. On lines 382 and 383 the authors point to a github repository:

<https://github.com/sysbiomelab/BioFung>

And claim that the database is shared through this repository. This repository only contains two bash scripts, but no data. Much later, under "Code availability" on line 337, the authors point to a web page with three files:

HGMA.web.MSP.abundance.matrix.csv

HGMA.web.metadata.csv

MGEM_models.zip

The README.md file of the github repository comes with a "usage" section, but this is very cryptic.

The first thing the authors must do is to make sure that the reader can access the database. What software dependencies are needed? Are there any limitations on versions of the software? Once installed, how to verify that the database is working. Finally, an example of searching the database for information could be included.

It is not possible to judge the merit of the database without supplying a way to use it and reproduce the authors results.

On line 100, the authors state that "The quality of the genome of each strain was checked at the level of scaffold assembly and genome size". What does this mean? Did some genomes fail this check?

The authors found that some pathways are enriched in the AGAu cluster. How were the genes representing the different pathways chosen (SPE11, SPE3, CKI1, TAZ1 and CEM1)? The choice of CEM1 is surprising, since it is not involved in fatty acid synthesis. Fatty acids are synthesized by the products of FAS1 and FAS2 genes, the CEM1 gene encodes a part of the mitochondrial fatty acid synthase that probably only produces lipoic acid for the lipolyation of mitochondrial enzymes. The authors should comment on this. Figure 4 is wrong in this respect, showing a direct connection between CEM1, fatty acids and triacylglycerol.

This work reports interesting observations, however, in my opinion, several issues must be addressed, before the manuscript meets the expected standards for publication in Communications Biology.

Dear reviewers,

We would like to thank you for the valuable comments on our manuscript entitled "**Integrative functional analysis uncovers metabolic differences between *Candida* species**" by Begum et al. Please find specific details of point-by-point reply to the reviewers' constructive comments and the changes we made to improve the findings of this manuscript.

Replies from reviewer 1:

Reviewer #1:

It is unclear whether the authors are trying to emphasize the new BioFung tool or the functional role of *Candida* species in humans.

In this manuscript we first focus on development of a path for better fungal functional annotation and development of the BioFung database. Our second focus was the application of the BioFung on *Candida* species to better understand their underlying metabolic functions and differences. We believe the BioFung database is part of the road to discovery of the importance fungal metabolism in host-fungal interactions as we provide a novel understanding of *Candida* metabolism with a focus on three specific *Candida* species (AGAu - - *C. albicans*, *C. glabrata* and *C. auris*). We provide meaningful and clinically relevant outcomes of *Candida* metabolism. We highlight the importance of amino acid metabolism in three *Candida* species that have been highly associated with candidaemia and mortality attributed to fungi. We demonstrated the effective use of BioFung with *Candida*, providing a novel understanding of metabolism with *Candida* species with no previous dissemination of genotypic/phenotypic similarity.

Reviewer #1:

The authors claim to have 128 fungal species in their database and developing the tool to elucidate the functional role of fungi; therefore, it would have been ideal to have shown the robustness of their tool using different fungal taxa, e.g., *Aspergillus*, *Malassezia* etc. and not just focus on one taxon *Candida* using 8 different species.

Thanks for your constructive comment. As it was suggested we performed additional analysis for *Aspergillus* strains to demonstrate the robustness and strength of the BioFung tool (Supplementary Figure 1c). We provide the output of core and accessory differences across three *Aspergillus* species and the distribution of level annotation of BioFung database (Supplementary Figure 1a-b). More specifically, KO coverage by Euk100 was approximately 25% less than BioFung (Extended figure 1c) with BioFung having yeast specific pathways that are not included in Euk100_v91 (Extended figure 1e). Comparison of BioFung directly with KEGG using *Aspergillus* showed there was 5% less metabolism annotation in KEGG organism (Supplementary figure1c).

Reviewer #1:

The authors should mention the limitations of their study and the benefits of their tool over other similar bioinformatics tools available as an open-source.

Thanks for the comment. Biofung is unique in providing functional annotation for fungi. This database is specialised in providing accurate and broader fungal-specific

KEGG ortholog annotation. Euk100.hmm (tool from Raven-Toolbox- <http://biomet-toolbox.chalmers.se/index.php?page=downtools-ravenm>) can provide KO annotation. We have compared BioFung to both these tools highlighting higher specificity to fungi (Extended Figure 1 and Supplementary Figure 1). However, BioFung is limited to the tools, databases and fungal genome annotation available. With limitations in fungal genome annotation and funding in expanding annotation for species variability there is significant scope for improving the BioFung database. BioFung is reliant on KEGG and the update versions on the 128 fungal species available, which has limited access based on KEGG license. Finally, BioFung, like all other similar databases, could have cross-annotation to human and other organisms' orthologs. This could be due to miss-annotation of KOs or a lack of specificity to individual fungal species as parallel fungal sequencing exists with no fungal specific function attached to the KOs. BioFung provides a unique tool in providing fungal annotation that can be paired with multi-omics data to divulge fungal capability despite these disadvantages.

Reviewer #1:

Overall, the manuscript needs and the results require detailed statistical analysis. We have gone through the entire manuscript again and added statistical analysis performed in brackets in relevant areas. We have also added statistical analysis in the figure legend and rescanned the manuscript to display clearer statistical analysis performed throughout this work within the methods. Finally, the reviewers mentioned the expansion of hypergeometric testing, which has been explained further in the methods as hypergeometric distribution using Wilcoxon testing to obtain P values.

Reviewer #1:

The methods section needs to be streamlined, and overall, the manuscript requires major revision in presenting the findings with emphasis on the BioFung tool. Thank you for bringing this matter into focus. In response to this comment, we have revised and reorganised the methods section and added a separate section to reflect the organised details of BioFung construction, quality assessment and application. We have brought to the fore the BioFung tool as a resource in exploring metabolism of fungal species and pinpointing metabolism of interest. This led us to explore the importance of metabolic diversity with metabolomics and gene expression confirming pathways that allow particular *Candida* species to dominate in mycobiome and cause disease.

Reviewer #1:

The references section needs to conform with the journal guidelines – the organisms names should be italicized, some text is in uppercase where not required, few references missing the information on Journal volume and page numbers. References were built based on an online tool which was laden with errors. We have manually checked the references and corrected them according to the guidelines, including the volume and page numbers. Greater care has been taken in italicising species names throughout the manuscript, including the reference.

Reviewer #1:

The abstract should describe full forms of the terminologies before using abbreviations .e.g., "KEGG", AGAu species".

Amendments have been made to the acronyms used in the abstract.

Replies from reviewer 2:

Reviewer #2:

It is not clear to me how the BioFung database is being made available to the readers. On lines 382 and 383 the authors point to a Github repository: <https://github.com/sysbiomelab/BioFung>. And claim that the database is shared through this repository. This repository only contains two bash scripts but no data. Much later, under "Code availability" on line 337, the authors point to a web page with three files: HGMA.web.MSP.abundance.matrix.csv, HGMA.web.metadata.csv, MGEM_models.zip. The README.md file of the Github repository comes with a "usage" section, but this is very cryptic. What are software dependencies needed? Are there any limitations on versions of the software? Once installed, how to verify that the database is working. Finally, an example of searching the database for information could be included. It is not possible to judge the merit of the database without supplying a way to use it and reproduce the authors' results.

A significant concern shown by both the reviewers was access to the BioFung database. This was an error on our part as BioFung was too large to include in the Github site. Thus, we redirected to the group's website, but there were unforeseen issues in uploading the tool we were unaware of and as a result it did not appear. We recognise the comments that the availability of BioFung was limited and elusive. This was due to the size of actual BioFung, and we have expanded on the "ReadMe" for better deployment of BioFung. We have corrected this by making BioFung available on Github under large-file-storage, which now can be directly downloaded from the Github repository: <https://github.com/sysbiomelab/BioFung>. BioFung requires Nextflow v21.04.1 and either singularity v3.8.3 or docker 20.10.7. All software executed by the pipeline is containerised meaning that no additional installation is required. HPC installation has ready dependencies for singularity/docker with clear instruction for Nextflow installation (<https://github.com/sysbiomelab/BioFung>). As suggested by reviewer, we have added an input example named afumigatis.fasta and example of output expected in ReadMe file. This was addressed under the code and data availability of the manuscript and changes made.

Reviewer #2:

On line 100, the authors state that "The quality of the genome of each strain was checked at the level of scaffold assembly and genome size". What does this mean? Did some genomes fail this check? Formal and rewrite.

we would like to thank the reviewer for the comment. On checking the quality of the genome, we indicate the variation of sequencing methodology, and the samples of *Candida* species that are available. This is important in the quality of functional

annotation that we provide with BioFung. We demonstrate that varying sequencing methods, assembly, and genome size may contribute to differences seen across the *Candida* species. This has been indicated within the manuscript for readers to be aware of the importance of quality and its impact on any form of annotation attached. The sentence was revised as below:

“Comparison of sequencing platform, scaffold assembly and genome were performed to assess quality of genome impacts the annotations”

Reviewer #2:

The authors found that some pathways are enriched in the AGAu cluster. How were the genes representing the different pathways chosen (SPE1, SPE3, CKI1, TAZ1 and CEM1)? The choice of CEM1 is surprising since it is not involved in the fatty acid synthesis. Fatty acids are synthesized by the products of FAS1 and FAS2 genes, the CEM1 gene encodes a part of the mitochondrial fatty acid synthase that probably only produces lipoic acid for the lipoylation of mitochondrial enzymes. The authors should comment on this. Figure 4 is wrong in this respect, showing a direct connection between CEM1, fatty acids and triacylglycerol.

We found the feedback from the reviewers very constructive in adding *FAS1* and *FAS2* analysis in the representation of fatty acid biosynthesis (please see updated Figure 4 and extended figure 4). We initially included *CEM1* as an ideal fatty acid biosynthesis gene for investigation as *Cem1* is a β -ketoacyl-ACP synthase in yeast mitochondrial fatty acid biosynthesis within type-II FAS system indicating a direct link between *Cem1* function and fatty acid metabolism [PMID: 16950653, PMID: 8412701, PMID: 17604452, <https://biocyc.org/gene?orgid=CALBI&id=ORF19.5977>,]. We have removed *CEM1* from our data and performed new experiments to include direct gene expression of fatty acid biosynthesis with *FAS1* and *FAS2*. The manuscript was revised based on these changes and the FAS experiments figure was added to Extended Fig. 4d.

Reviewer #2:

This manuscript is unfortunately laden with many small errors. This manuscript needs proofreading and the attention of someone proficient in English. I do recommend spelling the authors' names correctly. The Spelling errors that I found are marked in red in the pdf file.

Each of the errors has been corrected based on the attached document. Furthermore, greater care has been taken with the grammar and spelling of the manuscript, and we hope that it meets the standard.

Reviewers' comments:

Reviewer #2 (Remarks to the Author):

While there were substantial improvements in this manuscript, this paper continues to be quite opaque. One problem is the apparent confusion on what this paper is about? As the paper is both about a tool and an analysis of a number of strains, the paper is hard to read.

For example, what does the BioFung database do on a fundamental level? After reading the GitHub readme I assume that BioFung takes a list of protein coding sequences in FASTA format (representing a genome) and outputs KEGG orthologues. These orthologues are then used to annotate the genome covered by the FASTA sequences.

If I am correct, please add this to the manuscript.

There is a deeper problem with the text. The figures are very nice and clear for the most part. However, the authors do not aid the reader in the interpretation of the results. It is as if the figures and the text were made by two different people. The authors should aid the reader in a much closer way.

The authors define a group of three strains as AGAu species - *C. albicans*, *C. glabrata* and *C. auris*. Apparently these cluster together based on the KO terms they code for?

On line 143 the authors say "Additionally, xylan and sugar carbohydrate utilization were the dominant functions in the accessory genome". They then go on to say that "Cell wall composition is a crucial virulence factor".

This is surprising as xylan is only found in plant cell walls. Looking through the strain list, several famous D-xylose fermenters can be found: *C. maltosa*, *C. tenuis*, *C. intermedia* and *C. arabinofermentans*. The authors go on to state (line 172) that the AGAu cluster is enriched in functions associated to a lifestyle in a niche where wood biomass is broken down. The authors should at least comment on this surprising finding. The authors should perhaps double check this result to verify that there was no switch between AGAu and non-AGAu?

The authors state on line 192 that "We identified significant enrichment of amino acid metabolism, including arginine, proline, cysteine and methionine metabolism. We also observed significant levels of fatty acid biosynthesis and glutathione metabolism, which have previously been associated with virulence mechanisms". This presumably means that there is an enrichment of these KO in the virulent AGAu group compared to the non-AGAu?

Figure 3a show identical results for AGAu and non-AGAu. This is commented on line 186 as "we identified enriched pathways in AGAu species", however later on line 188 "as no differences were observed between cluster group in pathway analysis". Is this a contradiction? On line 192 "We identified significant enrichment of amino acid metabolism, including arginine, proline, cysteine and methionine metabolism." Enrichment of KO terms? In relation to what?

In Figure 3c, no significant difference is found in concentrations of triacylglycerol between non-AGAu and AGAu? Yet they are marked as such in Figure 4? I am not surprised that there are triacylglycerol produced by yeasts growing on sabouraud dextrose broth. Any yeast strain does this. I am also not surprised that genes that are essential in *S. cerevisiae* are expressed in all strains tested here.

Finally, the authors draw far reaching conclusions on in-vitro metabolomics experiments on media with little relevance to the metabolic situation during host infection. As the metabolism changes greatly depending on carbon source, the authors should be explicit about this limitation.

I believe that, given the issues mentioned above, the manuscript does not meet the expected standards for publication in Communications Biology.

Reply to Reviewer #2:

We would like to thank the reviewer 2 for pointing out the confusions that remain within the manuscript and editor for giving us an opportunity to address these issues.

1) While there were substantial improvements in this manuscript, this paper continues to be quite opaque. One problem is the apparent confusion on what this paper is about? As the paper is both about a tool and an analysis of a number of strains.

Thank you for the comment on the purpose of the manuscript. We would like to highlight that this manuscript addresses the issue of lack of functional analysis tool for fungal species and along with other tools we have demonstrated the power of the tool using *Candida* species. To remove any confusion about the purpose of this manuscript we have added a paragraph within the introduction (lines 84-91) to explain the usefulness of this work.

2) For example, what does the BioFung database do on a fundamental level? After reading the GitHub readme I assume that BioFung takes a list of protein coding sequences in FASTA format (representing a genome) and outputs KEGG orthologues. These orthologues are then used to annotate the genome covered by the FASTA sequences. If I am correct, please add this to the manuscript.

A simple explanation has been added to the introduction of the BioFung in the manuscript before divulging the overall makeup and function of the BioFung database.

3) There is a deeper problem with the text. The figures are very nice and clear for the most part. However, the authors do not aid the reader in the interpretation of the results. It is as if the figures and the text were made by two different people. The authors should aid the reader in a much closer way.

The figures and text were produced by the 1st author. Due to her dyslexia, her ability to represent data in figures is far stronger. Nevertheless, the authors have expanded the description within the text and highlighted in red throughout the manuscript.

4) The authors define a group of three strains as AGAu species - *C. albicans*, *C. glabrata* and *C. auris*. Apparently these cluster together based on the KO terms they code for?

The groupings were determined from literature review of *Candida* species contributing to candidemia and mortality and this has been now reflected in the text.

5) On line 143 the authors say "Additionally, xylan and sugar carbohydrate utilization were the dominant functions in the accessory genome". They then go on to say that "Cell wall composition is a crucial virulence factor".

This sentence has been rewritten as the reviewer correctly pointed out confusing sentences.

6) This is surprising as xylan is only found in plant cell walls. Looking through the strain list, several famous D-xylose fermenters can be found: *C. maltosa*, *C. tenuis*, *C. intermedia* and *C. arabinoferramentans*. The authors go on to state (line 172) that the AGAu cluster is enriched in functions associated to a lifestyle in a niche where wood biomass is broken down. The authors should at least comment on this surprising finding. The authors should perhaps double check this result to verify that there was no switch between AGAu and non-AGAu?

After double checking the analysis was definitely only performed on *C. albicans*, *C. glabrata* and *C. auris* (AGAu cluster). With odd ratios highlighting enrichment only in AGAu pathways. However, comment has been added to manuscript to reflect an explanation in both results and discussion sections.

7) The authors state on line 192 that "We identified significant enrichment of amino acid metabolism, including arginine, proline, cysteine and methionine metabolism. We also observed significant levels of fatty acid biosynthesis and glutathione metabolism, which have previously been associated with virulence mechanisms". This presumably means that there is an enrichment of these KO in the virulent AGAu group compared to the non-AGAu?

Thank you for highlighting this area of confusion, there was enrichment of these KO pathways in both groups. We attempted to display this across the cluster group, however as kindly pointed out by reviewer we have changed this figure 3a to reflect enrichment of these pathways in *Candida* species using Wilcoxon. This is also addressed in the next comment.

8) Figure 3a show identical results for AGAu and non_AGAu. This is commented on line 186 as "we identified enriched pathways in AGAu species", however later on line 188 "as no differences were observed between cluster group in pathway analysis". Is this a contradiction? On line 192 "We identified significant enrichment of amino acid metabolism, including arginine, proline, cysteine and methionine metabolism." Enrichment of KO terms? In relation to what?

This is related to the same comment as replied to before. The bar chart doesn't encompass the difference in KO from AGAu and non-AGAu. Changes have been made to figure 3a to reflect enrichment in *Candida* species. We have re-written this area to concentrate on similarities and differences that we are observing in metabolomics.

9) In Figure 3c, no significant difference is found in concentrations of triacylglycerol between non-AGAu and AGAu? Yet they are marked as such in Figure 4? I am not surprised that there are triacylglycerol produced by yeasts growing on sabouraud dextrose broth. Any yeast strain does this. I am also not surprised that genes that are essential in *S. cerevisiae* are expressed in all strains tested here.

Thank you for pointing out this potential confusion in Figure 3c and 4. The metabolomics data represented in Figure 3c is from species grown in SAB. Thus, although there will be triacylglyceride in SAB media, this will be the same for all fungal species investigated here, not just the AGAu cluster. Therefore, any differences in the levels will be as a result of fungal metabolism rather than differences in the medium. The differences between triacylglyceride in AGAu and non-AGAu is significant as determined by Wilcoxon testing ($P < 0.05$). This was an error of omission in the original figure and has been amended in the resubmitted manuscript.

10) Finally, the authors draw far reaching conclusions on in-vitro metabolomics experiments on media with little relevance to the metabolic situation during host infection. As the metabolism changes greatly depending on carbon source, the authors should be explicit about this limitation.

Our use of the in vitro metabolomics described in this study was to demonstrate that there are differences in the metabolic potential between different *Candida* species and to show how some of these differences were in pathways that are related to virulence during infection, indicating the potential of the approach we have taken to analysing more appropriate clinical metabolomics datasets in the future. This limitation has been added to line 322 onwards.